# Utilization of a Cortical Xenogeneic Membrane for Guided Bone Regeneration: A Retrospective Case Series

**DOI:** 10.3390/jcm13154575

**Published:** 2024-08-05

**Authors:** Cyril Debortoli, Arthur Falguiere, Fabrice Campana, Jean-Hugues Catherine, Delphine Tardivo, Romain Lan

**Affiliations:** 1Oral Surgery Department, Assistance Publique des Hôpitaux de Marseille, 264 Avenue St Pierre, 13005 Marseille, France; arthur.falguiere@laposte.net (A.F.); fabrice.campana@univ-amu.fr (F.C.); jean-hugues.catherine@univ-amu.fr (J.-H.C.); lanromain@live.fr (R.L.); 2Laboratory ISM, Aix-Marseille University, CNRS, EFS, 13005 Marseille, France; 3Laboratory ADES, Aix-Marseille University, CNRS, EFS, 13005 Marseille, France; delphine.tardivo@univ-amu.fr

**Keywords:** bone graft, guided bone regeneration, cortical lamina, mandible posterior, case series

## Abstract

**Background**: Guided bone regeneration (GBR) is a reliable technique used in vertical and horizontal bone defects. The posterior mandibular region is an area limited by anatomic constraints. The use of resorbable membranes with a cortical component could compensate for the lack of rigidity of resorbable membranes without the complications of non-resorbable membranes. The aim of this study was to evaluate the mean bone gains of a xenogeneic cortical membrane in horizontal and vertical bone defects in comparison with other membranes in the literature. **Methods**: A porcine cortical membrane was used to perform 7 GBR in the posterior mandibular region of five patients. Preoperative (T0) and six months postoperative (T1) cone beam computed tomography were superimposed to measure the horizontal and vertical bone gain. Implants were positioned at all sites, six months after GBR. Complications and bone resorption around the implants were also documented. **Results**: The mean horizontal and vertical bone gains were 3.83 ± 1.41 mm and 4.17 ± 1.86 mm, respectively. The analysis of repeatability was 0.997. As many as 40% of patients experienced pain refractory to analgesics. No exposure or infectious phenomenon was observed. **Conclusions**: This xenogeneic cortical membrane seemed to provide interesting results in the regeneration of horizontal and vertical bone defects. Comparative and prospective studies are necessary to validate the effectiveness of this membrane.

## 1. Introduction

GBR involves the use of a resorbable or non-resorbable membranous biological barrier that excludes epithelial and connective tissue cells, providing stability and maintaining the grafted space to promote the migration of stem cells capable of bone regeneration [1,2].

The “Sausage technique” is a GBR stabilized by pins, allowing better stabilization of the biomaterial and its retention in space relative to the flap; essential conditions for success [3]. This surgical technique avoids a second operative site and, depending on the type of membrane used, is indicated for horizontal and vertical bone reconstructions [1,3]. Resorbable membranes have the advantage of not requiring a second removal procedure and promoting mucosal healing, but their low rigidity restricts their role in maintaining space vertically. They are indicated for strictly horizontal bone defects, classified as Cawood and Howell Class IV [4]. In contrast, non-resorbable membranes have higher rigidity and are suitable for vertical or mixed bone defects, classified as Class V or VI, but may have higher rates of exposure, complications, and less ease of manipulation [4,5].

GBR in the posterior mandibular sector poses a surgical challenge due to its vascularization deficiency, essential for bone regeneration, and the presence of vital anatomical elements that can complicate the surgical procedure. But also, muscular constraints from the cheek, floor, and tongue exert deleterious tensions and stresses on bone grafts [6,7].

However, as an alternative to short implants or more invasive bone graft techniques (surgical distraction, onlay graft), guided bone regenerations (GBR) are still increasingly used in posterior mandibular reconstructions despite the surgical difficulties [1].

Lamina^®^, marketed by Osteobiol^®^ by Tecnoss^®^, is a resorbable membrane composed of collagen and bone frameworks, exhibiting physicochemical properties with higher rigidity, tensile strength, and tear resistance compared to other resorbable collagen membranes. These characteristics could offer better stability and protection for bone grafts, especially vertically, also reducing exposure complications through collagen treatment and slow resorption optimizing the migration and coverage capacity by epithelial cells [8]. These membranes could expand the indications and success of resorbable membranes, without the complications relating to non-resorbable membranes. However, there are few articles on the use of these membranes in the literature. Initially used for orbital floor fracture reconstructions [9,10,11], Lamina^®^ was first reported intraorally in 2010, showing average vertical and horizontal gains of 2.5 mm and 3.7 mm, respectively, in four patients undergoing horizontal bone augmentation [12].

Complications of bone grafts are mainly suture dehiscence, membrane exposure, and infectious complications. In the posterior mandibular region, management of the inferior dental nerve can have consequences such as paresthesia or hypoesthesia [13]. Based on Fontana’s classification [14], Gallo et al. have proposed a decision tree for non-absorbable membranes that can also be applied to absorbable membranes [15].

The objective of this study was to evaluate, at six months postoperatively, horizontal and vertical bone gains using Lamina^®^ with the sausage technique on partially edentulous patients with Cawood and Howell Class IV to VI bone defects in the posterior mandibular region.

## 2. Materials and Methods

A retrospective case series analyzing pre- and postoperative bone levels was conducted on patients who underwent GBR using the sausage technique with Lamina^®^ in the posterior mandibular region between September 2022 and January 2023, conducted by a senior surgeon. The primary objective was to assess vertical and horizontal bone gains (in millimeters) six months after guided bone regeneration using Lamina^®^ (Osteobiol^®^ by Tecnoss^®^). The protocol was approved by the Ethics Committee of Assistance Publique des Hôpitaux de Marseille (PADS23-80), and the study was conducted in accordance with the Declaration of Helsinki.

Patients requiring posterior mandibular bone reconstruction for implant placement, capable of understanding and signing an informed consent form, aged 18 years or older, and presenting Cawood and Howell Class IV, V, or VI bone defects were considered for inclusion [4].

The exclusion criteria included patients: (i) with a smoking addiction of more than 10 cigarettes per day, (ii) under immunosuppressive or immunodepressive medications, (iii) with uncontrolled systemic diseases, (iv) receiving systemic corticosteroids, (v) at risk of jaw osteonecrosis (received or receiving bisphosphate treatment/radiated in the head and neck region with more than 30 Gy), (vi) undergoing chemotherapy outside the head and neck region in the last two months before surgery, (vii) with alcohol or drug dependence, (viii) with psychiatric disorders, (ix) with untreated periodontal disease or AAP/EFP Classification stage III to IV periodontitis (2018) [16], and (x) pregnant or lactating women.

Patients without postoperative follow-up or lost to follow-up were excluded from the study.

The primary outcome was the difference in height and thickness of the bone ridge on the operated site between T1 (6 months postoperatively) and T0 (preoperative measurement). Evaluation was performed using three-dimensional radiological examination by cone beam computed tomography (CBCT) at T0 and T1, performed by the same Planmeca^®^ ProMax 3D Mid unit. CBCT data were registered in Planmeca Romexis^®^ software (version 6.2.1.25) and transferred to Mimics^®^ software (version 23.0) for superimposing T0 and T1 CBCT images. CBCTs were performed by the same operator, with consistent settings: field dimensions of 10 cm × 6 cm, configuration 668 × 668 × 401 pixels, and a slice thickness of 150 μm, with settings of 90 kV, 13.0 mA, and 1057 mGy/cm². To ensure optimal measurement reproducibility at both operative times, the CBCTs were computer superimposed using at least 6 bone or dental landmarks selected based on the operative site (Table 1 and Figure 1). Dental landmarks, subject to modification over time, were used only after verifying that no anatomical or radiological changes had occurred.

A panoramic curve was generated on the superimposed result to enable measurements on the same implant sections. All measurements were performed by another operator (CD, different to the surgeon) for each future implant site, ensuring measurement repeatability by conducting them initially (R0) and at 1 month (R1). The measurement protocol (Figure 2) involved:▪A line perpendicular to the top of the preoperative ridge (red, named D1);▪A line passing through the most coronal and apical points of the preoperative ridge (orange, named D2);▪Lines D1 and D2 intersect at a point named A;▪Point B is defined as equidistant from the most coronal and apical points of the preoperative ridge on line D2;▪Horizontal measurement 1 (HM1): from point A to the most anterior point of the postoperative ridge (dark green);▪Horizontal measurement 2 (HM2): from point B to the most anterior point of the postoperative ridge (light green/yellow);▪Vertical measurement (VM): from the midpoint of MH1, named C, to the most coronal point of the postoperative ridge (dark blue).

For each clinical case, each measurement was recorded three times by the same operator (CD) for each measuring time (R0 and R1), systematically generating three final average measurements: a vertical average measurement, corresponding to vertical bone gain relative to the most coronal plane of the preoperative bone ridge, and two horizontal average measurements corresponding to horizontal bone gain relative to the most coronal plane of the preoperative bone ridge. The two horizontal measurements evaluated the average horizontal bone gain in relation to the body and neck of a future implant with ideal positioning, knowing that the thickness of the buccal bone is crucial for the survival of the implant.

The repeatability of the measurements was measured using the Lin concordance coefficient.

All interventions followed the same predefined surgical protocol (Figure 3): A crestal and intrasulcular incision extending up to two teeth on each side of the area of interest. A full-thickness flap exposed the bone defect. A semi-thickness incision was made to obtain flap laxity. Lamina^®^ Standard 30 × 30 mm was hydrated for 5 min. Pins or osteosynthesis screws were placed lingually to stabilize the membrane, and the particulate xenogeneic bone material of porcine origin (Creos^®^—Nobel^®^) mixed with autologous bone at a 50/50 ratio was placed on the grafting area. The membrane was then stabilized on the vestibular wall (minimum of 2 screws or osteosynthesis screws on the vestibular and lingual sides). Mattress sutures apical with absorbable suture thread Vicryl ^®^ 4/0 to the mucogingival line were realized over the bone defect. A combination of continuous and simple non-absorbable suture thread Prolène ^®^ 5/0 was used to achieve primary closure.

The postoperative prescription included amoxicillin 1000 mg (2 tablets per day for 7 days) or, in case of allergy, clindamycin 300 mg (2 tablets in the morning and evening for 7 days). Prednisolone 20 mg (1 mg/kg for 3 days), tramadol 100 mg LP (1 capsule in the morning and evening), and paracetamol 1000 mg (1 tablet every 6 h), supplemented this prescription. A chlorhexidine mouthwash was performed from 48 h postoperatively (twice daily for 10 days), along with a surgical toothbrush.

Implants (of different brands) were positioned at all sites at least six months after GBR. No immediate implant loading was carried out. The definitive implant-supported prosthetic rehabilitations were carried out at least 3 months after implant placement.

Second outcomes focused on postoperative complications after GBR noted during the first 6 months of follow-up (dichotomous response):▪The presence of pain not relieved by the use of Tier I and II analgesics during the one-week postoperative follow-up appointment;▪Membrane exposure;▪The presence of infectious phenomena related to the operative site (site suppuration);▪The presence of nerve complications—hypoesthesia of the inferior alveolar nerve.

Furthermore, the implant success rate and bone resorption around implant loading one year after their placement were also noted:▪The implant success rate was evaluated according to—clinical absence of inflammation of the peri-implant mucosa, absence of periodontal pocketing greater than 2 mm upon peri-implant probing, and implants in function;▪Bone resorption was assessed by 2D intraoral radiographs on the bone level in relation to the implant neck at 3 months and 1 year after implant placement.

The total follow-up after GBR was therefore one and a half years. This case series has been reported in line with the PROCESS guidelines [17].

## 3. Results

Five patients aged 45 to 62 years were included (3 females, 2 males). Seven Cawood and Howell Class IV to VI bone defects in the posterior mandible were operated on. The mean horizontal and vertical bone gains were 3.83 ± 1.41 mm and 4.17 ± 1.86 mm, respectively (Table 2).

Regarding the measurements, the analysis of repeatability according to the Lin concordance coefficient showed repeatability equal to 0.997 for the operator (Table 3).

Evaluation of second outcome:Two out of five patients (40%) experienced postoperative pain at the one-week postoperative follow-up appointment;No patient experienced membrane exposure, infectious phenomena, or nerve complications;The implant success rate was 100%;Radiographic analysis around the loaded dental implants at 3-month and 1-year follow-ups reported, respectively, 0.2 mm and 0.44 mm of peri-implant bone loss (Table 4);

The results the first and second outcome are summarized in Table 5.

## 4. Discussion

This case series reported an average horizontal and vertical bone increase of 3.83 ± 1.41 mm and 4.17 ± 1.86 mm, respectively. Despite the small size of our series and therefore the impossibility of comparisons with other more robust, comparative or prospective studies, these results seemed to correspond with those found in the literature:▪In the horizontal direction, for all membranes combined, Sanz-Sanchez et al. [18] (2015) reported a mean horizontal gain of 3.90 mm (95% CI, 3.52–4.28 mm), and Elnayef et al. (2018) a gain of 3.61 ± 0.27 mm [19]. With resorbable collagen membranes only, Wessing et al. (2018) found lower results (2.27 ± 1.68 mm) among 460 patients [20].▪In the vertical direction, for all membranes combined, Urban et al. (2019) [21] reported a similar mean gain of 4.18 mm, whereas Wessing et al. (2018) showed a mean vertical gain of 3.05 ± 1.02 mm with resorbable membranes only [20].▪Focusing on the posterior mandibular sector, for all membranes combined, our results approached those of Elnayef et al.‘s systematic review, which reported a mean vertical gain of 3.83 ± 0.49 mm from a sample of 62 patients [19]. With non-resorbable membranes only, Robert et al. (2023) showed, in their systematic review, a mean vertical gain of 4.7 mm (minimum: 1.5 mm–maximum: 5.24 mm) [22].

The results found in this series appear to be more similar to those of non-resorbable membranes, suggesting the possible use of Lamina^®^ membranes for vertical defects. However, these literature results should also be interpreted cautiously due to the lack of clear methodology provided by each selected article, without distinguishing between operative sites [22].

The results in this case series were also consistent with those of other studies using Lamina^®^. For horizontal defects, Foti et al. found an average horizontal increase of 4.17 mm (min: 2.50 mm; max: 5.55 mm) in a series of 5 patients, while a unique clinical case reported by Rossi et al. showed a horizontal gain of 6.8 mm. The vertical bone gain, the only parameter evaluated by Rossi et al., was reported at 2.55 mm, much lower than our results [23,24,25]. In our study, five out of the seven sites operated on were associated with major vertical defects, where the pursuit of horizontal gain was minimal compared to vertical gain. This then may explain the slightly lower horizontal gain in our results. Comparing results across different sites (maxillary/mandibular, anterior/posterior) must be nuanced. The posterior mandibular sector differs from other oral sectors due to the presence of a powerful muscle belt that can exert pressure and tension on the bone graft. Mammoto and Stucker’s teams have shown that the presence of pressure and tension has a detrimental role in neo-angiogenesis, potentially compromising the success of bone grafts [6,26]. The presence of the inferior dental canal also limits the procedure due to the possibility of post-surgical nerve complications.

This work showed the presence of postoperative pain in 40% of cases. There does not seem to be any biological explanation or in the composition of these membranes that could justify these results. Perhaps, in this small case series, questions may arise regarding the adherence to analgesic treatments. Another factor can be the semi thickness incision leading to bleeding and then postoperative edema.

This case series showed no membrane exposure or infectious signs.

Peri-implant bone resorption was assessed on 2D intra-alveolar radiographs. The implant success rate and the reduced bone loss at 1 year of follow-up seems to indicate good bone stability over time and satisfactory osseointegration for the implants, but Serino and al. showed that the radiographical measurements underestimated peri-implant bone loss [27]. This step shows a limitation of the study due to the imprecision of these radiographs. Moreover, the measurement of marginal bone loss makes it possible to identify local and general factors in implant health. Güven et al. showed that local factors had more significant effects on MBL than did systemic factors; therefore, regular monitoring is necessary [28].

According to Derks et al.’s work, marginal bone loss greater than 0.5 mm associated with bleeding or suppuration on peri-implant probing is associated with peri-implantitis. In our case, there were no problems in terms of probing at implant sites n°5 and 6. Galindo-Moreno and al. established the cut-off value of 0.5 mm of bone loss at 6 months and a radiographical level of 2 mm of bone loss defined as moderate/severe peri-implantitis is the value up to which success or survival can be defined. Marginal bone loss could therefore be linked to other possible local factors (torque, implant positioning, prosthetic abutments, and soft tissue) [28,29].

A single operator performed the surgeries, limiting biases in preoperative judgment; however, concerns about the accessibility and manipulation of this biomaterial still remain.

The main limitations of this case series were the number of patients and the difficulty in superimposing CBCTs to achieve the most accurate measurement on the same CBCT section pre- and postoperatively. Andriola et al., based on their literature review, estimated a margin of error for measurements after superposition ranging from 0.01 to 0.26 mm [30].

To minimize this margin of error, the authors decided that one operator would take three measurements at R0 and R1 to obtain a more accurate estimate of horizontal and vertical bone gain. The repeatability was excellent and validated the measurement technique in this study.

Artificial intelligence could be used to set landmarks automatically and more quickly, but the literature shows that an experienced specialist is needed to check these, as there can be a margin of error of 2 mm [31]. Hendrickx et al. showed that generalizability and robustness need further improvement in terms of accuracy and efficiency to rival experienced specialists [32].

However, the quantity of bone gain, due to bone distortion and the size of the voxel, could be overestimated [33]. To improve measurement precision and optimize the measurement protocol, another operator would be needed to calculate intra- and inter-operator variability and minimize the margin of error.

In order to adhere to the “As Low As Reasonably Achievable” (ALARA) principle, the use of mid-field CBCT resulted in the absence of skeletal points usually used in the literature [34,35,36].

As compensation, dental points were used to obtain a maximum number of available landmarks and optimize superposition precision. The precision of dental points has been validated in previous studies compared to rounded bone structures (porion, condyle, etc.) [34,37,38].

To optimize precision, bone points were chosen in accordance with the literature data, with structures close to the median sagittal plane and non-curvilinear [13,23,25]. Regarding this biomaterial, Lamina^®^ membranes have two xenogenic origins: equine or porcine. Porcine origin biomaterials have an antigenic structure close to humans, favorable porosity, and a sintering surface state that seems to allow slower resorption and minimized inflammatory reaction. Salamanca et al. [39], Bracey and al. [40], Seo and al. [41] showed that the structure of porcine cancellous bone, its microstructure, and porosity were similar to human bone.

The physico-chemical characteristics studied by Caballé-Serrano et al. showed higher tensile strength (2.1 MPa) in porcine-origin Lamina^®^ compared to other membranes on the market (all below 1 MPa). Porcine-origin Lamina^®^ also seems to have a higher Young’s modulus in dry and hydrated states (4.8 and 0.9 MPa, respectively) indicating higher rigidity (less than 2 and 0.5 MPa, respectively, for other membranes). Its ability to absorb, when hydrated, is approximately three times lower than other membranes, making them easier to handle with a reduced risk of tearing [8].

According to the results described in this series, these membranes seem to have indications in the reconstruction of horizontal, vertical, and/or mixed bone defects in the posterior mandibular region.

## 5. Conclusions

GBR with Lamina^®^ in the posterior mandibular sector, seemingly combining the advantages of resorbable and non-resorbable membranes, have reported similar results to those found in the literature, whether in the horizontal direction with resorbable membranes or in the vertical and/or horizontal direction with non-resorbable membranes. However, due to the small number of cases in our study and the absence of a control group, comparison with the literature data must be very careful and nuanced. The low rate of complications found in this case series is encouraging compared to non-resorbable membranes, as is the absence of a second operative site compared to autologous grafts, reducing intervention comorbidity. This biomaterial could find its application in the rehabilitation of Cawood and Howell Class IV to VI bone defects in the posterior mandibular sector, using its physico-chemical properties and resorbable capacity.

## Figures and Tables

**Figure 1 jcm-13-04575-f001:**
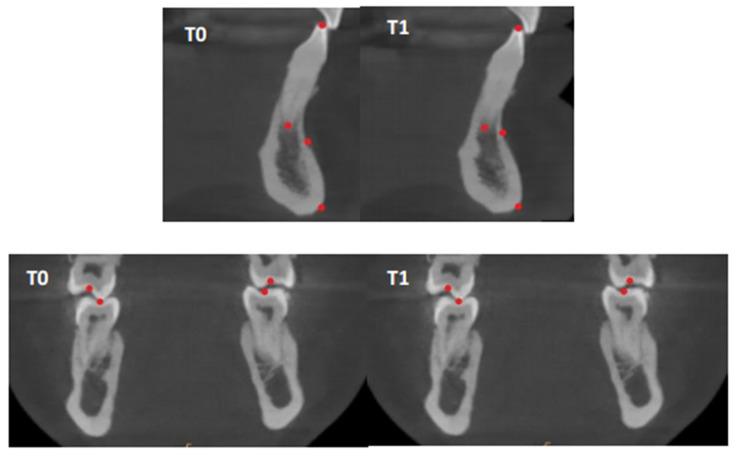
Example of setting up landmarks on three-dimensional sections using Mimics^®^ software (version 23.0) at T0 and T1. Landmarks used are: point B; point Menton; apex of mandibular incisors; free edges of mandibular incisors; maxillary and mandibular molar central fossa.

**Figure 2 jcm-13-04575-f002:**
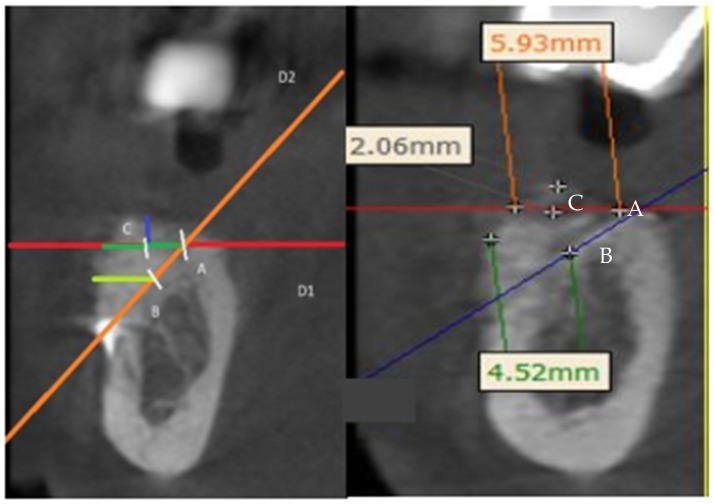
Protocol for horizontal and vertical measurements. Line D1 (red): line perpendicular to the top of the preoperative ridge; Line D2 (orange): line passing through the most coronal and apical points of the preoperative ridge; Lines D1 and D2 intersect at a point named A; Point B is defined as equidistant from the most coronal and apical points of the preoperative ridge on line D2; point C is used to calculate the vertical measurement: from the mid-point of horizontal measurement 1 (from point A to the most anterior point of the postoperative ridge (dark green)) to the most coronal point of the postoperative ridge (dark blue).

**Figure 3 jcm-13-04575-f003:**
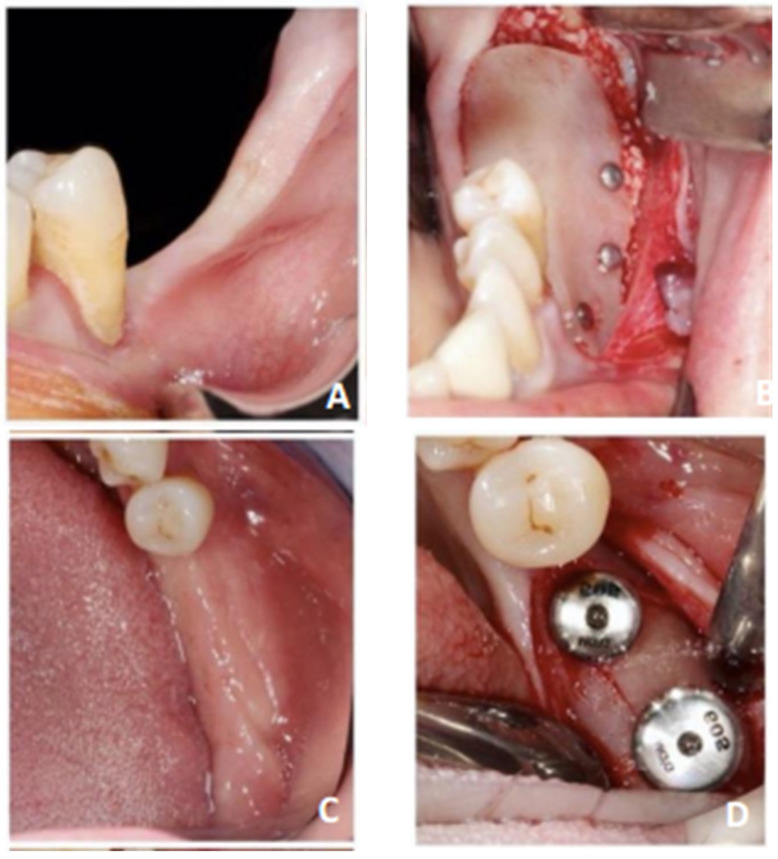
Summary of surgical steps illustrated in patient n° 4. (**A**) Preoperative view. (**B**) After placement of the cortical lamina^®^. (**C**) Six months post-surgery. (**D**) Re-opening and pose of two implants. A membrane residue is visible. (**E**) CBCT preoperative. (**F**) CBCT postoperative at 6 months. (**G**) A 30 × 30 mm Lamina^®^ cortical membrane.

**Table 1 jcm-13-04575-t001:** Definition and description of landmarks used.

Landmarks	Description
**BONE**	
Point B	Most inferior point on the midline of the image of the anterior concavity of the mandible
Point Menton	Most medial point and lowest part of the mandible
**DENTAL**	
Apices of tooth 41, 31	Most apical point on part of the tooth
Free edges 42, 41, 31,32	Most coronal point on part of the incisors on a sagittal section
Central fossae 36, 46	Point most apical part of the cuspid fossa

**Table 2 jcm-13-04575-t002:** Representation of mean horizontal and vertical gains in millimeters based on times R0 and R1.

		Measure Time		General Mean (±SD)
		R0	R1	
Horizontal gain (mm)	Mean HM1	3.84	3.85	3.84 ± 1.05
	Mean HM2	3.88	3.73	3.81 ± 0.90
	Mean HM1 + HM2	3.86	3.79	3.83 ± 1.41
Vertical gain (mm)	Mean VM	4.14	4.20	4.17 ± 1.86

HM1: horizontal measure 1. HM2: horizontal measure 2. Mean HM: mean of horizontal measures. VM: vertical measure. R0: first measurement. R1: second measurement, 1 month after first measurement. SD: standard deviation.

**Table 3 jcm-13-04575-t003:** Measurement of intra-operator repeatability according to the Lin concordance coefficient for operator CD.

Types of Measure	Lin Concordance Coefficient between R0 and R1
HM1	0.998
HM2	0.996
VM	0.998
Mean	0.997

HM1: horizontal measure 1. HM2: horizontal measure 2. VM: vertical measure. R0: first measurement. R1: second measurement, 1 month after first measurement. Mean: retained value.

**Table 4 jcm-13-04575-t004:** Peri-implant bone loss (in millimeters) at T1 (3 months post operative) and T2 (1 year post operative).

	Peri-Implant Bone Loss (mm)	
Implants Sites	T1	T2
1	0.1	0.3
2	0	0.4
3	0.3	0.4
4	0.2	0.2
5	0.4	0.8
6	0.3	0.6
7	0.1	0.4
**Mean peri-implant bone loss (mm)**	0.2	0.44

**Table 5 jcm-13-04575-t005:** Summary of horizontal and vertical gains and complications in the case series.

Patient	Cawood and Howell	Horizontal Mean Gain	Vertical Mean Gain	Pain	Membrane Exposure	Infection	Nerve Trouble	Failure
1	V	5.11	3.93	Y	N	N	N	N
2	V	5.25	1.42	Y	N	N	N	N
	V	4.3	2.57					
3	VI	1.26	9.35	N	N	N	N	N
4	IV	3.56	3.32	N	N	N	N	N
5	V	3.65	4.05	N	N	N	N	N
	V	3.66	4.35					

Y: Yes. N: No.

## Data Availability

For further details or raw measurement data, please contact the corresponding author.

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
