# Peer review of "Utilization of a Cortical Xenogeneic Membrane for Guided Bone Regeneration: A Retrospective Case Series"

_jcm, 2024, doi:10.3390/jcm13154575_

Round 1

Reviewer 1 Report

Comments and Suggestions for Authors

Utilization of a cortical xenogeneic membrane for guided bone 2 regeneration: a retrospective series.

The aim of the present investigation was to assess the use of resorbable membranes with a cortical component for bone augmentation.

GENERAL COMMENTS

The article is in-line with the journal topic, but flaws should be improved.  The study is interesting.

There are many spelling mistakes

Abstract

Background: The scope of the study should be declared in this section.

Introduction

1.      This section is very short, please add more information on the complication around Bone lamina and bone augmentation.

2.      ….How-65 ever, there are few articles on the use of these membranes in the literature….

Kindle reading the follow papers: PMID: 30574770, PMID: 30126102, PMID: 27469552 etc.

Materials and methods

1.      Only 5 patients was evaluated ?

Results

Kindly add the corss section before and after bone augmentation

Images are of poor quality, please remove them and reproduce them in high resolution.

Discussion

Please add the limit the study.

Author Response

Comments 1 :

Background: The scope of the study should be declared in this section.

Response 1 : Added

Comments 2 : This section is very short, please add more information on the complication around Bone lamina and bone augmentation.

Response 2 : Done. 

But there is no articles which mentionned bone lamina complication. We cited Fontana and Gallo about the management of complications. 

Comments 3 : How-65 ever, there are few articles on the use of these membranes in the literature….

Kindle reading the follow papers: PMID: 30574770, PMID: 30126102, PMID: 27469552 etc.

Response 3 : 

  • PMID: 30574770, PMID: 30126102 : These papers show the use of lamina in sinus bone grafting, which is not the subject of this article.
  • PMID : 27469552 : This case series does not calculate average bone gain or describe how bone gain is calculated, does not use CBCT overlays, and does not discuss possible complications.

These articles have not been included in our introduction because they are of no interest and are not related to our subject.

Materials and methods

Comments 4 : Only 5 patients was evaluated ?

Response 4 : 5 patients with 7 implant sites.

Results

Comments 5 : Kindly add the corss section before and after bone augmentation

Response 5 : Done

Comments 6 : Images are of poor quality, please remove them and reproduce them in high resolution.

Response 6 : Done

Discussion

Comments 7 : Please add the limit the study.
Response 7 : 

We have mentioned in the discussion the various limitations of this work

- the small number of patients

- CBCT superimposition

- the absence of a reproducibility test with inter-operator variability

  • the use of two-dimensional X-rays to measure marginal bone loss

Thanks for yours comments ! 

Reviewer 2 Report

Comments and Suggestions for Authors

The paper is considered to be significant in terms of content. However, there is a lack of explanation of the materials and methods used, which should be corrected.

(1) There is a lack of explanation about the membrane (Lamina), which is an important point of this paper. Please describe in detail, including ingredients, and add Fig.

(ii) The explanation of the measurement methods for VM, HM1 and HM2 is very difficult to understand. Please explain in detail using enlarged images.

(iii) The reason for measuring R1 and R2 is unclear.

(iv) Please describe the reason for MBL after implant placement in the discussion.

Author Response

The paper is considered to be significant in terms of content. However, there is a lack of explanation of the materials and methods used, which should be corrected.

Comments 1 : There is a lack of explanation about the membrane (Lamina), which is an important point of this paper. Please describe in detail, including ingredients, and add Fig.

Responses 1 : Added in Fig. Ingredients about these membranes are kept confidentials despite our attempts to contact the referring laboratory.

Comments 2 : The explanation of the measurement methods for VM, HM1 and HM2 is very difficult to understand. Please explain in detail using enlarged images.

Responses 2 : Corrected. VM is measured  from the midpoint of MH1 (not D1 as first version), named C, to the most coronal point of the postoperative crest (dark blue). The two horizontal measurements are used to measure bone growth at the implant neck and implant body. The horizontal increase is therefore taken as the average of the two. In our view, this method of calculation is more consistent. Many articles on bone grafting do not explain how they are measured.

With this protocol, we feel that we are measuring reliably (with repeatability measurements) and more accurately. As stated in the article, another examiner could be used to improve reproducibility, something that will be improved in a study.

Comments 3 : The reason for measuring R1 and R2 is unclear.

Reponses 3 : To validate our CBCT measurement and overlay protocol, we decided to measure at time R0 and then at one month (R1) to check the operator's repeatability. We wanted to optimize and verify the reliability of our measurement protocol.

Comments 4 : Please describe the reason for MBL after implant placement in the discussion.

Responses 4 : Done

Thanks for yours comments ! 

Reviewer 3 Report

Comments and Suggestions for Authors

Upon reviewing the article "Utilization of a cortical xenogeneic membrane for guided bone regeneration: a retrospective series," several areas of concern and potential errors were identified: The study reports postoperative pain in 40% of cases but lacks a clear biological explanation for this occurrence. This could suggest either an issue with the adherence to analgesic treatments or other factors not considered in the study​. The use of 2D intra-alveolar radiographs to assess peri-implant bone resorption might have limitations, as noted by Serino et al., who indicated that radiographic measurements could underestimate bone loss. This represents a significant limitation due to the imprecision of these radiographs​. The small number of patients and the challenges in superimposing CBCT scans for accurate measurements are highlighted as limitations. These factors introduce a margin of error in measurements, ranging from 0.01 to 0.26 mm, which could affect the study's conclusions. The study acknowledges that the bone gain measurements might be overestimated due to bone distortion and voxel size. To improve accuracy, it suggests involving another operator to assess intra- and inter-operator variability, which was not done in this case​. To adhere to the ALARA principle, mid-field CBCT was used, which lacks skeletal points typically referenced in literature. Instead, dental points were utilized, which could introduce variability and affect the precision of superimposition​.There are concerns about the accessibility and manipulation of the biomaterial used, which were not thoroughly addressed. This could influence the practical application and reproducibility of the results​. The tables feature some diagonal lines that are quite unusual for a scientific publication. Could these be modified to conform to the format of international publications? The references are appropriate, though some are somewhat dated (1988-2012); if possible, I suggest reviewing more recent literature. These points suggest areas for improvement in future studies and highlight the importance of addressing these limitations to strengthen the findings and their clinical applicability. Despite these concerns, the innovative approach and potential clinical benefits presented in this study provide a strong foundation for further research and validation.

Author Response

Upon reviewing the article "Utilization of a cortical xenogeneic membrane for guided bone regeneration: a retrospective series," several areas of concern and potential errors were identified:  

Comments 1 : The study reports postoperative pain in 40% of cases but lacks a clear biological explanation for this occurrence. This could suggest either an issue with the adherence to analgesic treatments or other factors not considered in the study​.

Responses 1 : It’s mentioned in the discussion section. Another factor can be the semi thickness incision because it’s lead to a bleeding and then post operative It will be added in the discussion.

Comments 2 : The use of 2D intra-alveolar radiographs to assess peri-implant bone resorption might have limitations, as noted by Serino et al., who indicated that radiographic measurements could underestimate bone loss. This represents a significant limitation due to the imprecision of these radiographs​.

Responses 2 : Mentioned in the discussion section as you said

Comments 3 : The small number of patients and the challenges in superimposing CBCT scans for accurate measurements are highlighted as limitations. These factors introduce a margin of error in measurements, ranging from 0.01 to 0.26 mm, which could affect the study's conclusions. The study acknowledges that the bone gain measurements might be overestimated due to bone distortion and voxel size. To improve accuracy, it suggests involving another operator to assess intra- and inter-operator variability, which was not done in this case​.

Responses 3 : Mentioned in the discussion section. The team is working on it to improve future bone graft to measure repeatability and reproducibility.

Comments 4 : To adhere to the ALARA principle, mid-field CBCT was used, which lacks skeletal points typically referenced in literature. Instead, dental points were utilized, which could introduce variability and affect the precision of superimposition​.

Responses 4 : Mentioned in the discussion section. Moreover, some dental points are more accurate than some bone landmarks as mentioned.

Comments 5 : There are concerns about the accessibility and manipulation of the biomaterial used, which were not thoroughly addressed. This could influence the practical application and reproducibility of the results​.

Responses 5 : The surgical protocol was predefined and cited in the materials and methods section. `The biomaterial was hydrated during five minutes and fixed with pins or screws.

Comments 6 : The tables feature some diagonal lines that are quite unusual for a scientific publication. Could these be modified to conform to the format of international publications?

Responses 6 : Modifications done

Comments 7 : The references are appropriate, though some are somewhat dated (1988-2012); if possible, I suggest reviewing more recent literature.

Responses 7 : The reference dated from 1988 is the classification of Cawood and Howell. It’s the most utilized classification in bone defects.

- References from 2012 is about cbct landmarks which are defined on the basis for landmark positioning

Comments 8 : These points suggest areas for improvement in future studies and highlight the importance of addressing these limitations to strengthen the findings and their clinical applicability. Despite these concerns, the innovative approach and potential clinical benefits presented in this study provide a strong foundation for further research and validation.

Responses 8 : These points will be improved in larger future studies.

Thanks for yours comments ! 

Round 2

Reviewer 2 Report

Comments and Suggestions for Authors

The points raised have been appropriately corrected.

It is a clear and understandable paper that can be accepted.